# CFRP U-Wraps and Spike Anchors for Enhancing the Flexural Performance of CFRP-Plated RC Beams

**DOI:** 10.3390/polym15071621

**Published:** 2023-03-24

**Authors:** Jamal A. Abdalla, Haya H. Mhanna, Alnadher B. Ali, Rami A. Hawileh

**Affiliations:** Department of Civil Engineering, American University of Sharjah, Sharjah P.O. Box 26666, United Arab Emirates

**Keywords:** carbon-fiber-reinforced polymers, reinforced concrete, flexural strengthening, U-wraps, FRP spike anchors

## Abstract

Deterioration of infrastructure is a major challenge in the civil engineering industry. One of the methods that has been deemed effective in upgrading reinforced concrete (RC) structures is using externally bonded fiber-reinforced polymer (FRP) composites. However, the efficacy of this retrofitting technique is limited by the premature debonding failure of the FRP at the concrete-FRP interface; thus, the full capacity of the FRP is rarely utilized. Anchorage systems were proposed as a feasible solution to suppress or delay debonding failure. This paper presents an experimental investigation on the use of end U-wraps and carbon FRP (CFRP) spike anchors to anchor CFRP plates bonded to flexure-deficient RC beams. The experimental program consisted of seven RC beams with the length of the CFRP plate, type of anchorage, and the number of anchors as experimental variables. Test results indicated a remarkable enhancement in the ultimate load-carrying capacity when longer CFRP plates were used to strengthen the beams. In addition, anchoring the plates enhanced the strength of the CFRP-plated beams by 16–35% compared to the unanchored specimen, depending on the anchorage type and scheme. Finally, fib Bulletin 90 (2019) provisions provided the most accurate predictions of the moment capacity of the strengthened specimens.

## 1. Introduction

Recently there has been an escalating need to restore and lengthen the serviceable duration of deteriorated structures. The causes of the deterioration of civil infrastructure are due to various factors such as corrosion of steel, excessive loading, and aging [1,2,3]. In the last few decades, a novel technology has emerged using fiber-reinforced polymers (FRPs) to restore and enhance reinforced concrete (RC) structures. The special interest in using FRPs in the retrofitting industry is due to their many favorable characteristics, such as excellent corrosion resistance, high strength-to-weight ratios, and durability [4,5,6,7]. FRPs have various forms (sheets, plates, and bars) that are utilized depending on the structural application. Externally bonding FRP plates to the beam’s soffit is commonly used to enhance the flexural capacity of RC beams; such beams are referred to as FRP-plated beams [8,9,10,11,12]. In practice, the imperfection of the workmanship in placing externally bonded FRP sheets and plates may result in defects and consequently affect the performance of the beam. Wan et al. 2018, conducted experimental tests and studied the effect of defects in externally bonded FRP-reinforced concrete under flexure [13].

FRP-plated beams often fail in a brittle manner by debonding of the plates from the beam’s soffit at low FRP strain levels. The debonding failure mainly occurs in two forms: (1) concrete cover separation, which initiates at the plate end and propagates along the tension steel bars towards the center of the beam leading to detachment of the FRP with a thick layer of concrete from the beam’s soffit, or (2) intermediate crack debonding which initiates at the bottom of the flexural cracks in the high moment region and propagates towards the FRP plate end [14,15,16,17]. In general, debonding failure limits the full potential of the FRP system as it occurs at FRP strain levels lower than the rupture strain of the FRP (around 30–40%) [18]. One of the methods that was proven to be effective in delaying and mitigating debonding failure is anchoring the FRP plates. The main function of the anchors is to resist interfacial slip at the plate end and/or counteract the shear stresses at the bond interface, thus providing load continuity between the FRP and concrete after local debonding [10]. Accordingly, the strain in the FRP plates would be utilized, resulting in substantial increases in the load-carrying capacity of the beams [19,20,21,22].

Different types of anchorage systems were developed and proposed up to date. Earlier studies examined mechanical anchorage, which involves using metallic or nonmetallic bolts, angles, or anchor rods to anchor the FRP into the concrete. A study by Lamanna et al. [23] investigated the effect of strengthening RC beams with mechanically fastened RC strips. Test variables included FRP elastic modulus, predrilling, and fastener lengths and layouts. Experimental results showed that the inclusion of mechanical fasteners resulted in an enhancement in the stiffness and post-yield stiffness of the strengthened beams. In addition, the anchored beams gained higher ductility and failed in a progressive manner similar to that of the unstrengthened control beam [23]. Similarly, Bank et al. [24] conducted tests on beams strengthened with a combination of carbon FRP (CFRP) and glass FRP (GFRP) strips anchored with continuous strand mats that were fastened to the concrete with steel powder-actuated fasteners and expansion anchors. Test results indicated that the yield and ultimate loads were improved by 25 and 58% compared to the unstrengthened beam, respectively. The advantage of the anchors was also displayed in the significant enhancement in the ductility of the strengthened specimens [24]. In another study, Wu and Huang [25] used mechanical fasteners composed of a thin steel plate and two small nails to anchor the FRP laminates. It was concluded that the inclusion of the mechanical fasteners resulted in improving the interfacial bond strength more than seven times that of the unanchored FRP sheets. It was also concluded that the bond strength increased with the increase in the number of fasteners and embedment of the nails [25]. Despite its advantages, the use of mechanical anchorage is limited by several factors such as high cost, low durability, vulnerability to corrosion, labor intensiveness, and impracticality [26]. In addition, commercially available FRP do not have sufficient longitudinal bearing and open hole strengths to allow the transfer of forces to the FRP by bearing [24].

More recently, several studies investigated FRP U-wraps as an anchorage system to mitigate FRP debonding failure [14,15,16,27,28]. The application of the U-wraps involves placing the FRP sheets in a U-shape form perpendicular to the beam’s longitudinal axis. The U-jackets are generally located at either both ends of the flexural FRP sheet or equidistant throughout the beam, although the former is more popular. The purpose of the U-wraps is to resist the tensile stresses at the end of the FRP sheet and to suppress crack propagations at the end of the FRP and/or throughout the beam span [14,26]. A study by Tahsiri et al. [28] showed that anchoring the CFRP laminates with end CFRP U-wraps resulted in a 6–30% enhancement in the load-carrying capacity of the RC beams. In addition, the debonding failure was delayed, which resulted in higher beam ductility and energy dissipation capacity. However, a significant reduction in the strength was depicted at the onset of debonding after which the load-deflection curve progressed similar to that of the unstrengthened control beam [28]. In another study by Al-Tamimi et al. [27], the authors investigated the effect of different CFRP plate lengths and U-wrap anchor configurations on the flexural capacity of RC beams. It was concluded that increasing the CFRP plate length increased the load-carrying capacity of the specimens. In addition, depending on the plate length, the double layers of U-wraps improved the capacity of the RC specimens by 14–26% compared to single U-wraps. However, the double U-wrap layers significantly reduced the specimens’ ductility [27]. In a recent research investigation, Fu et al. [15] studied the effect of inclined U-wraps on the performance of FRP-plated beams. It was reported that the inclined U-jackets shifted the failure mode from intermediate crack debonding to the more favorable concrete crushing and FRP rupture. In addition, using 45° U-wrap inclination angle enhanced the load-carrying capacity by 20–56% depending on the width and height of the U-wraps. The performance of the beams increased with the increasing width and height of the U-wraps. In addition, the 45° U-wraps resulted in higher beam ductility and higher FRP strain utilization. In contrast, the 90° and 135° U-jackets did not aid in a significant enhancement in the load-carrying capacity and FRP strain utilization. Besides mechanical anchors, several researchers explored different methods of increasing the bond of externally bonded FRP systems. Jiang et al., 2018 [29] proposed an epoxy interlocking approach to enhance FRP-to-concrete bond, whereas Sanginabadi et al., 2022 [30] proposed groove anchorage methods that enhanced the performance of strengthened beams. 

Another anchorage system that was proven effective in attaining the full capacity of the FRP is FRP spike anchors. FRP spike anchors are made of a bundle of fibers soaked in epoxy resin, with one end of the fibers inserted into the concrete and the other end splayed in a fan shape over the FRP plate [31,32,33]. The main advantage of FRP spike anchors, as opposed to other anchorage systems, is their applicability to a wide range of structural applications and their compatibility with the FRP sheets [34,35,36]. A number of studies investigated the use of FRP spike anchors in flexural [37,38,39] and shear [40,41,42,43,44] applications and under direct pullout [45,46,47]. The focus has been on the effect of different anchor parameters and anchor alignment on the strength enhancement of the FRP system. In general, the preceding literature indicated that the use of FRP spikes in strengthening RC structures has beneficial effects on the overall performance of the strengthened RC beams. Particularly, the strength and ductility of the anchored specimens were significantly upgraded compared to the unanchored ones. Sun et al. [48] conducted an experimental study to investigate the parameters that affect the strength of the anchors. The authors studied the effect of concrete strength, anchor fan angle, bonding of the CFRP strip, and anchor-to-sheet material ratio on the capacity of concrete prisms strengthened with CFRP sheets and spike anchors. Several conclusions were drawn from this study. First, an anchor-to-sheet material ratio of 2 and a fan angle of less than 63° are recommended to reduce stress concentration and develop the tensile strength of the FRP. In addition, increasing the concrete strength enhanced the capacity of the FRP anchors. Finally, bonding the CFRP strip to the concrete resulted in a higher ultimate load and prevented premature anchor rupture [48]. Zaki et al. [37] tested five full-scale T-beams to study the flexural strength enhancement provided by different arrangements of FRP spike anchors. The inclusion of anchors resulted in 95–172% strength enhancement compared to the control beam. In addition, closely spaced anchors along the shear span were more effective than widely spaced anchors. Finally, using side FRP spike anchors did not contribute to delaying the debonding as the failure occurred midspan before the propagation of debonding reached the end anchor [37].

In light of the above background, the main aim of this research is to study the effect of various anchorage techniques, namely, end U-wraps and CFRP spike anchors, on the flexural behavior of CFRP-plated beams. Seven RC beams were tested under four-point bending tests, with the length of the CFRP plate, type of anchor, and the number of CFRP spike anchors as the test variables. The influence of these parameters was examined in terms of the failure modes, load capacities, CFRP strain, and ductility of the RC beams. In addition, the experimental results were compared with the ACI440.2R-17 [49] predictions. 

## 2. Materials and Methods

### 2.1. Specimen Design

The experimental program consisted of seven RC beams with similar geometry composed of a length, width, and height of 1840, 120, and 240 mm, respectively. The effective depth from the compression fiber was 202 mm. The flexural reinforcement consisted of two φ10 mm steel bars in the tension zone and two φ8 mm steel bars in the compression zone. In addition, the beams were reinforced with φ8 mm rectangular stirrups spaced at 80 mm c/c to avoid shear failure. The geometry and reinforcement details of the RC beam specimens are shown in Figure 1.

Among the seven beams, one beam was left unstrengthened to serve as a control benchmark specimen (CB). Two specimens were strengthened with unanchored CFRP plates of different lengths to study the effect of CFRP plate length on the capacity of the specimens. Particularly, specimen S was strengthened with a 1 m long CFRP plate at the soffit midspan of the beam (Figure 2a), and specimen F was strengthened with a 1.69 m long CFRP plate that was bonded at the beam’s soffit between the supports (Figure 2b). The width of the CFRP plate (100 mm) was kept constant among all tested beams. All anchored specimens were strengthened with 1 m long CFRP plates, as shown in Figure 2. Specimen S-U was anchored with 200 mm wide CFRP U-wraps on both ends designed to mitigate debonding failure, as demonstrated in Figure 2c.

The rest of the specimens were strengthened with different numbers and configurations of CFRP spike anchors. Particularly, specimens S-2A, S-4A(d), and S-4A(s) were strengthened with two spike anchors (one on each plate end), four distributed spike anchors, and four spike anchors two of which were located on each side of the plate ends, respectively. All other anchor details were kept constant for all beams to determine the impact of the number of anchors and anchor arrangement on the capacity of the beams. Particularly, the beams were drilled with a 10 mm hole diameter at an anchor embedment depth of 80 mm and an insertion angle of 90°, as shown in Figure 3. In addition, the anchor diameter, fan length, and splay width were 8 mm, 120 mm, and 100 mm for all beams except specimen S-4A(s) which had anchors with splay widths of 50 mm, as shown in Figure 3c. The strengthening details of all the CFRP-spike anchored beams are presented in Figure 3. In addition, the test matrix of the tested beams is summarized in Table 1.

### 2.2. Material Properties

All beam specimens were cast with self-consolidating concrete (SCC) with a designed compressive strength of 45 MPa and a water-to-cement (w/c) ratio of 0.36. A total of sixteen standard concrete cylinders with a length of 200 mm and diameter of 100 mm were also cast to determine the mechanical properties of the concrete. All cylinders were tested during the beams’ testing week, eleven of which were tested under compression in accordance with ASTM C39/C39M-18 [50], and five were tested in tension (splitting test) in accordance with ASTM C496/C496M-11 [51]. The mix proportions of the concrete and the average obtained compressive and tensile strengths of concrete are shown in Table 2.

Steel bars of 8 and 10 mm diameter were used to reinforce the beams in flexure and shear. To determine the mechanical properties of the steel bars, three samples of both sizes were tested under tension in a universal testing machine (UTM) in accordance with ASTM A370-18 [52] standard. The average yield stress (*f_y_*), strain at yield (*ε_y_*), ultimate strength (*f_u_*), and modulus of elasticity (*E_s_*) of No. 8 steel bars were 618 ± 10.7 MPa, 0.0035 mm/mm, 683 ± 4.3 MPa, and 193 ± 2 GPa, respectively. In addition, the *f_y_*, *ε_y_*, *f_u_*, and *E_s_* of No. 10 steel bars were 621 ± 17 MPa, 0.0041 mm/mm, 731 ± 15.8 MPa, and 182 ± 8 GPa, respectively.

Two types of CFRP materials were used in this study. The beams were externally reinforced with Sika CarboDur [53] plates, a carbon fiber plate used for structural strengthening applications. In addition, the U-wraps and spike anchors consisted of SikaWrap-300C [54] unidirectional carbon fiber fabric. The CFRP plates were bonded onto the RC beams using Sikadur 30 LP [55] adhesive and the U-wraps and spike anchors were impregnated and bonded to the beams with Sikadur-330 [56] two-part epoxy adhesive. The mechanical properties of the CFRP plates, CFRP sheets, and epoxy adhesives are summarized in Table 3. The source of CFRP plates, and sheets as well as the epoxy used in this study, is Sika GCC, UAE branch (https://gcc.sika.com/en/sika-gcc/sika-uae.html, accessed on 15 July 2017).

### 2.3. Installation Process

The surface of the beam specimens was prepared using a mechanical grinder with a steel disk to remove laitance and to provide a better bond with the CFRP plates. Next, the location of the CFRP plates and the holes were marked, and the 80 mm deep holes were drilled at the desired locations with a 90° angle and 10 mm diameter drill bits. Following that, the surface and the holes were cleaned to remove any dust particles. CFRP plates of width 100 mm were then cut at lengths of 1000 mm and drilled with openings at the anchor locations. In addition, two CFRP sheets with a width of 200 mm and a length of 600 mm were cut to be applied in the form of U-wraps onto specimen S-U.

CFRP spike anchors were prepared by cutting 60 × 200 mm pieces of SikaWrap-300C CFRP sheets. Each sheet was further cut into 6 pieces with widths of 10 mm and then regrouped at the key portion of the anchor using steel wires forming anchors with 80 mm embedment depth and 120 mm fan splay length. The diameter of the prepared anchors was then measured to ensure an 8 mm anchor diameter.

After preparing the plates and anchors, Sikadur 30 LP epoxy was mixed in the ratio of 3:1 by weight as provided by the manufacturer. Then, a layer of epoxy was placed onto the surface of the beams followed by the CFRP plates. Constant pressure was applied on the plates by hand for 2–3 min to ensure a strong bond between the CFRP and concrete. Finally, a layer of epoxy was applied to the plates. The anchor installation procedure started with mixing the low viscous Sikadur-330 epoxy adhesive with a ratio of 4:1 by weight, as specified by the manufacturer. For specimen S-U, the concrete surface was saturated with epoxy at the location of the U-wraps. Then, the impregnated U-wraps were applied at both ends of the CFRP plate using grooved rollers to remove entrapped air. Alternatively, for specimens with spike anchors, the holes were injected with epoxy, and the impregnated anchors were inserted. The fan portion was then splayed across the width of the CFRP plate. The excess adhesive was removed, and the epoxy was allowed to cure for one week before testing the beams.

### 2.4. Test Setup and Instrumentation

The test setup is illustrated in Figure 4. All beam specimens were simply supported with a clear span of 1690 mm and a shear span of 565 mm. A universal testing machine (UTM) that has a maximum static loading capacity of 1200 kN was used to apply the load at an increment of 10 kN per minute. The beams were tested under four-point bending tests where the load was applied midspan onto a steel spreader beam that distributed the load equally between the two loading points. All tests were terminated when the ultimate load dropped by 20%.

## 3. Results and Discussion

### 3.1. Load-Deflection Curves and Failure Modes

The applied load-deflection response curves of the tested beams are demonstrated in Figure 5. In addition, the observed failure modes are shown in Figure 6. The load-deflection curve of specimen CB shows a trilinear response which is typical for under-reinforced concrete beams. The first portion shows a region with high stiffness corresponding to the pre-cracking stage. Post-cracking, the stiffness of the load-deflection curve of specimen CB decreased due to the formation of the flexural cracks observed in the bending zone (Figure 6a). The formation of flexural cracks initiated at an applied load of 25 kN, as shown in Figure 5a. A sharp decrease in the slope of the load-deflection curve was depicted at the onset of the yield load (53 kN), after which the load increased gradually until attaining a peak load of 63 kN. After this point, crushing of concrete was visible at the top compression zone in the vicinity of the beam’s mid-span (Figure 6a) and the load-carrying capacity of the specimen declined until failure. The control beam achieved a ductile behavior beyond the yield point until the ultimate state.

It is clearly indicated in Figure 5 that the load-deflection curve of all the strengthened specimens showed an enhanced performance in terms of post-cracking stiffness, yield load, and peak load compared to the control beam CB. Specimen S, which was strengthened with a 1 m long CFRP plate, attained a peak load of 65 kN after which the slope of the load-deflection curve became flatter indicating a loss of flexural stiffness. Failure of specimen S was initiated by the formation of the shear-flexural cracks and vertical flexural cracks that spread over the middle span of the beam causing the attached CFRP plate to debond in a staged pattern. The debonding of the CFRP plate along with a thick concrete cover was followed by concrete crushing in the compression zone, as demonstrated in Figure 6b. On the other hand, the stiffness of the load-deflection curve of specimen F which was strengthened with a full-length CFRP plate did not experience a drop after the yield load (118 kN). In fact, the stiffness stayed constant, and the load increased until it reached an ultimate load of 126 kN at a much lower displacement than specimen CB, as shown in Figure 5a. A sudden drop was depicted after the ultimate state indicating the failure of the specimen due to the debonding of the CFRP plate from the concrete substrate (Figure 6c). It should be noted that very minor flexural cracks were observed during the testing of specimen F, indicating the advantage of strengthening in suppressing crack formation.

The load-deflection curve of specimen S-U that was anchored with end U-wraps propagated similarly to specimen S but with comparatively higher yield and peak loads. Particularly, the yield and peak loads of specimen S-U were 81 and 88 kN, respectively. The advantage of anchorage was also pronounced in the deflection of the beam, where the yield and peak loads occurred at higher deflections (5.3 and 6.9 mm, respectively) than that of specimen S (3.5 and 4.7 mm, respectively), as indicated in Figure 5b. After the ultimate state, a loss in the load-deflection capacity was widely observed due to the local debonding of the CFRP plates. However, the presence of anchorage resulted in sustaining the load for some time before failure. Specimen S-U ultimately failed due to a rupture in one of the U-wraps that was followed by concrete crushing in the compression zone between the loading points, as shown in Figure 6d.

It can be seen from Figure 5c that the load-deflection response of specimens S-2A, S-4A(d), and S-4A(s) that were strengthened with CFRP plates and spike anchors have increased post-cracking stiffness, yield load, and ultimate loads compared to the control specimen CB. The effect of anchorage is demonstrated in the enhanced load-carrying capacity of the anchored specimens compared to the unanchored specimen S. In addition, the load was sustained in the anchored specimens after the initial drop that occurred due to local debonding of the CFRP plate. This is because the anchors transferred the load from the CFRP plate to the concrete after local debonding.

Specimen S-2A experienced two sudden drops in the load-deflection curve; the first decrease which occurred at the onset of the peak load was due to the local debonding of the CFRP plate. After this stage, the load was sustained due to the engagement of the anchors, but the flexural-shear cracks increased and propagated from the end of the CFRP plate along the internal steel reinforcement. The second drop in the load-deflection curve of specimen S-2A corresponded to the point at which the anchors lost full tensile connection with the concrete. This action resulted in anchor pullout and complete debonding of the CFRP plate with attached concrete cover, as shown in Figure 6e. It should be noted that the enhancement in the capacity was at the expense of ductility, where specimen S-2A failed at a lower deflection (16.9 mm) than the unanchored specimen S (22.8 mm).

The stages of the load-deflection responses of specimens S-4A(d) and S-4A(s) were typical to S-2A, with the difference that S-4A(s) performed inferior to both S-2A and S-4A(d) in terms of capacity. The main failure mode of specimens S-4A(d) and S-4A(s) was debonding of the CFRP plate along with a thick concrete layer and anchor pullout, as shown in Figure 6f and g. The debonding in specimen S-4A(d) was followed by concrete crushing as depicted in Figure 6f. Comparisons between the anchored specimens will be discussed in depth in the next sections. The main findings of all the tested beams are summarized in Table 4. Ductility at the ultimate stage is defined as mid-span deflection at the ultimate stage divided by mid-span deflection at the yield state, while ductility at the failure stage is defined as mid-span deflection at the failure stage divided by mid-span deflection at the yield state.

### 3.2. Critical Loads

Figure 7a,b shows the critical loads (yield and peak loads) and the corresponding percentage increase with respect to the control beams CB and S. It can be seen from Figure 7a that the CFRP-plated beams have increased yield load compared to the control specimen CB. The enhancement in the yield load in the strengthened specimens mainly depended on the strengthening scheme, i.e., length of the CFRP plate, type of anchor, and the number of anchors. Specimen F, which was strengthened with a full-length CFRP plate, attained a superior yield load with 123% enhancement compared to the control beam CB. On the other hand, the load at which the steel yielded in specimen S, which was strengthened with a 1 m long CFRP plate, was 12% more than in specimen CB. Specimen F also attained remarkably higher ultimate load-carrying capacity than specimen S by 94%. It is observed from Table 4 that specimen S-4A(s), with anchors at each side (ends of the 1.0 m CFRP plate), failed at a lower load (Pu = 75.2 kN) than specimen S-4A(d), with anchors distributed along the length of the 1.0 m CFRP plate, that failed at a higher load (Pu = 84.2 kN). This is due to the relatively shallow embedment depth of the spike anchor and its small diameter.

The results indicate that the length of the CFRP plate has a significant effect on the performance of the strengthened beams, where the longer plate performed superior to the short-length plate. This behavior is attributed to the increased development length in the longer CFRP plate that resulted in enhancing the bond capacity between the CFRP plate and the concrete. According to the ACI440.2R-17 [49], the length of the FRP plate at the points between the cracking moment and ultimate moment for simply supported beams should be greater than the development length $\left(l_{df}\right)$ calculated by Equation (1). Consequently, the calculated $l_{df}$ is 174 mm and the lengths provided by the long and short CFRP plates were 565 and 220 mm, respectively. Although both lengths are greater than the calculated $l_{df}$, the bonded length in the long CFRP plate was more than double that of the short plate. Thus, the effective FRP stress developed at the section was greater in the longer CFRP plate and consequently improved the load-carrying capacity of the beam.
(1)$$l_{df}=\sqrt{\frac{nE_{f}t_{f}}{\sqrt{{f^{\prime }}_{c}}}}$$
where n is the number of CFRP layers; $E_{f}$ is the elastic modulus of the FRP (MPa); $t_{f}$ is the thickness of the FRP plate (mm); and ${f^{\prime }}_{c}$ is the compressive strength of concrete (MPa).

The difference in the performance of specimens S and F could be also contributed to the speed at which the flexural-shear cracks initiate from the end of the CFRP plate. The shorter plate caused increased stress concentrations at the termination points, resulting in faster crack initiation and propagation through the concrete, and eventually separation of the concrete cover. On the other hand, the longer CFRP plate debonded after the formation of flexural cracks at the pure bending span that propagated to the end of the plate. This resulted in more CFRP strain utilization and consequently better flexural performance.

With respect to the anchored specimens, specimen S-U which was anchored with end CFRP U-wraps provided the best performance in terms of enhancing the yield and peak loads. Overall, the inclusion of the U-wrap end anchorage system resulted in 36 and 35% improvement in the yield and peak loads as opposed to the unanchored specimen S, respectively. This distinct enhancement is due to the confinement provided by the U-wraps at the end of the CFRP plates that resisted the propagation of cracks from the CFRP plate ends. The U-wraps acted in prolonging debonding failure by maintaining the CFRP plates attached to the beams. Consequently, more CFRP strain was utilized resulting in increased load-carrying capacity of the beam.

The specimens anchored with CFRP spike anchors experienced 3–32% and 16–29% strength enhancement in the yield and peak loads over the unanchored specimen S, respectively. The results show that anchoring the CFRP plates with spike anchors is an effective method for delaying debonding failure and improving the beams’ capacity. However, the results of this group of specimens were not consistent. For example, using two anchors was more efficient than four anchors (two on each side). This could be because drilling two holes next to each other in the CFRP plate has caused possible fiber damage and matrix smearing in the vicinity of the holes. As a result, failure was initiated at these locations resulting in a reduction in the strength enhancement. On the other hand, four distributed anchors provided the best anchor configuration and significantly enhanced the load-carrying capacity of the beam (29% improvement with respect to specimen S). It should be noted that all the anchors failed by anchor pullout failure mode. The pullout capacity of the anchors could have been enhanced by providing longer embedment depths than 80 mm. In addition, the efficiency of the anchors could have been improved by using larger anchor diameters to increase the tensile capacity of the anchors.

### 3.3. Deflection and Ductility

The deflection at yield, ultimate, and failure loads and the ductility indices at ultimate and failure are provided in Table 4. It is evident from Table 4 that the CFRP-plated beams attained significantly lower mid-span deflection values at the critical loads than the unstrengthened control beam. Particularly, the reduction in the deflection at ultimate and failure loads ranged from 58–85% and 25–76% with respect to the control beam, respectively. This is attributed to the premature debonding of the CFRP plate which occurs at low CFRP strain levels.

Figure 8 shows the percentage reduction in the ductility of the tested beams at ultimate and failure states with respect to the unstrengthened control beam CB. It can be indicated from Figure 8 that the load enhancement depicted in the CFRP-plated beams was at the expense of ductility. In particular, the ductility of all CFRP-plated beams at ultimate was significantly inferior to that of the control beam by 31–81%. Specimen F showed the poorest ductility performance among all specimens with 81% and 83% reduction at ultimate and failure loads, respectively. On the other hand, specimen S which was strengthened with a short-length CFRP plate displayed a ductile behavior after the ultimate state (similar to that of the control beam). This occurred possibly due to the immediate debonding failure of the long CFRP plate after reaching the peak load which caused a sudden drop in the load-deflection response. However, the debonding of the concrete cover in the short-length CFRP plate initiated at the onset of the peak load, after which cracks formed in the concrete and the beam eventually failed by CFRP plate debonding followed by concrete crushing. This behavior resulted in a ductile response before failure.

Anchoring the CFRP plates did not improve the ductility at the ultimate state. Particularly, the ductility at the ultimate of the anchored specimens was reduced by 31–79% compared to the control beam. Ductility results also showed that the anchorage systems adopted in the study (i.e., end U-wraps and CFRP spike anchors) did not have a positive effect on the ductility at failure load (compared to the unanchored specimen S). This is because the anchored specimens attained higher peak loads than specimen S after which the specimens failed due to the U-wraps rupture or anchor pullout without much enhancement in the ductility. Specimens S-U and S-2A showed similar ductility responses, as shown in Figure 8. However, increasing the number of CFRP spike anchors slightly enhanced the ductility. Particularly, the best ductility performance was depicted in specimen S-4A(s) which was strengthened with four side spike anchors. This specimen had a comparable ductility response with the control beam at the failure state. This shows that the presence of two anchors on each side of the CFRP plate effectively improves the ductility performance of the beams by delaying failure. It is worth noting that the ductile behavior of the beams could be improved by providing more anchors along the length of the CFRP plate and by increasing the anchor embedment depth to enhance the pullout capacity of the anchors.

## 4. Analytical Predictions

The flexural moment capacity of the tested FRP-plated beams was predicted using the recent versions of the ACI440.2R-17 [45], CSA-S806.12 (R 2017) [57], and fib Bulletin 90 (2019) [58] design provisions. Table 5 summarizes the design equations used to calculate the nominal flexural strength of the RC beam specimens. In all design provisions, the bending moment of the strengthened cross section is calculated based on principles of RC design. The neutral axis depth is first calculated from strain compatibility (strains in the FRP, steel, and concrete are directly proportional to their distance from the neutral axis) and internal force equilibrium. Then, the design moment is obtained by moment equilibrium.

The following assumptions are suggested by the design codes to calculate the moment resistance of the strengthened cross-section: (1) the slip between the FRP and concrete is negligible; (2) the FRP has linear elastic behavior until rupture; (3) the shear deformation within the adhesive layer is neglected; (4) the tensile strength of concrete is neglected; (5) the surface preparation of the concrete substrate is sufficient to achieve the level of bond strength required in the design; (6) the member is assumed to be fully cracked at ultimate state; and (7) the extreme compression strains in concrete at failure as per the ACI440.2R-17 [49], CSA-S806.12 (R 2017) [57], and fib Bulletin 90 (2019) [58] design provisions are 0.003, 0.0035, and 0.0035, respectively.

Fib Bulletin 90 (2019) [58] provides several approaches to calculate the effective strain in the FRP based on the failure mode. The typical failure modes of FRP-strengthened beams include intermediate crack (IC) debonding, end debonding, and concrete cover separation. For the IC debonding, two methods are described in fib Bulletin 90 (2019) [58] to calculate the effective stress in the FRP. The first method is the simplified FRP stress limit method which assumes that the tensile stress in the FRP reinforcement does not exceed the lower fractile (5%) value of the bond strength at the ultimate state. The second method is the more detailed method which requires extensive calculations as it is based on determining the crack spacing and checking the force in the FRP between two adjacent cracks. On the other hand, the end debonding analysis for interfacial debonding proposed by fib Bulletin 90 (2019) [58] is conducted on the basis of FRP anchorage capacity. The two approaches consider the FRP curtailment point; however, one method does not take the position of flexural cracks into consideration, whereas the other method considers the position of the flexural cracks so that the end debonding analysis is conducted at the flexural crack closest to the point of zero moment or at an arbitrary concrete element between cracks. Finally, the end debonding analysis for concrete cover separation considers the additional vertical offset between the steel stirrups and the longitudinal FRP strip which builds up the tensile forces between the internal reinforcement and the FRP. The acting forces can be determined approximately with a truss model. In this study, the simplified FRP stress limit method, shown in Table 5, was opted to calculate the effective strain in the FRP. This method was chosen due to its simplicity and more conservative strain predictions.

Table 6 summarizes the results in terms of the ultimate load obtained from the tests (*P_exp_*), predicted load (*P_pred_*), and the ratio of *P_exp_* to *P_pred_*. It should be noted that all strength reduction factors were set to unity to allow for true comparison between different design standards. It is evident from Table 6 that all the design standards overestimate the capacity of the CFRP-plated beams. Particularly, the average ratios of *P_exp_* to *P_pred_* calculated as per the ACI440.2R-17 [49], CSA-S806.12 (R 2017) [57], and fib Bulletin 90 (2019) [58] design provisions are 0.55, 0.56, and 0.80, respectively. Fib Bulletin 90 [58] provisions provided safe and conservative predictions for specimen F with a full-length CFRP plate. However, the equations tend to overestimate the capacity of the unanchored and anchored short-length CFRP plates (the ratio of *P_exp_* to *P_pred_* ranged from 0.61–0.82).

The ACI440.2R-17 [49] and CSA-S806.12 (R 2017) [57] equations resulted in very close predicted load values (156 and 154 kN, respectively). This is because both standards use the same equation for the maximum tensile strain in the FRP due to debonding which governed the strain limit. In general, the ACI440.2R-17 [49] and CSA-S806.12 (R 2017) [57] provided unsafe and inaccurate predictions for all the strengthened specimens. The ratio of *P_exp_* to *P_pred_* ranged from 0.42 to 0.82. Specimen S which was strengthened with a short-length CFRP plate attained an ultimate load that was significantly less than the predicted load (percentage difference of 82%), whereas the predictions of specimen F were close to the experimental result (percentage difference of 21%). The equations provided by the ACI440.2R-17 [49] and CSA-S806.12 (R 2017) [57] do not account for the length of the FRP plate. As a result, the predicted capacities for specimens S and F were similar, whereas the experimental load value of the latter was substantially higher. It is recommended that future versions of the design standards incorporate the length of the FRP sheet in determining the flexural moment capacity of the section.

With respect to the anchored specimens, all design standards overestimated the capacity of the RC beams strengthened with anchored CFRP plates irrespective of the type of anchor (U-wrap or spike anchors). It should be noted that fib Bulletin 90 (2019) [58] provides an equation to estimate the capacity of anchored specimens. This equation is based on the efficiency of the anchorage system which should be confirmed by testing to comply with the design assumptions. Therefore, it was not used in this study to predict the capacity of anchored specimens. The ACI440.2R-17 [49] and CSA-S806.12 (R 2017) [57] do not provide a stress limit for the anchored specimens. Hence, the predicted results were typical for anchored and unanchored CFRP-plated beams for all the design standards. The future recommendation includes providing design equations for different types of anchors in the design codes to calculate the corresponding effective strain in the FRP based on the properties and geometry of the anchors.

## 5. Conclusions

This paper presents the results of an experimental study on the flexural behavior of CFRP-plated RC beams. The experimental program involved testing seven RC beams: one specimen was left unstrengthened (control beam), two were strengthened with unanchored short-length and full-length CFRP plates, and the rest of the beams were strengthened with short-length CFRP plates that were anchored with either U-wraps or CFRP spike anchors. All specimens were tested under two-point loading tests, and the performances of the strengthened specimens were compared to that of the control beam. The following conclusions can be drawn from the test results of the study:Strengthening RC beams using CFRP plates enhanced the yield and peak loads by 12–123% and 3–99% compared to the control beam, respectively.The length of the CFRP plate has a significant effect on the performance of the strengthened beams, where the longer plate improved the ultimate load-carrying capacity of specimen F by 94% compared to the short-length plate (specimen S). However, the load enhancement of specimen F was at the expense of ductility, which was reduced by 83% at failure load compared to specimen S.The inclusion of U-wraps to anchor the CFRP-plates prolonged the debonding failure and enhanced the yield and peak loads by 36 and 35% as opposed to the unanchored specimen S, respectively.Anchoring the specimens with CFRP spike anchors was beneficial in delaying the debonding of the CFRP plates and resulted in a 16–29% enhancement in the ultimate load-carrying capacity of the specimens over the unanchored specimen.Four distributed CFRP spike anchors provided the best anchor configuration and significantly enhanced the load-carrying capacity and ductility of the beam.Anchoring the CFRP plates did not improve the ductility at ultimate and failure states. However, increasing the number of CFRP spike anchors slightly enhanced the ductility. For better ductile performance, it is recommended to provide more anchors along the length of the CFRP plate and to increase the anchor embedment depth.The ACI440.2R-17 and CSA-S806.12 (R 2017) provided unsafe and inaccurate predictions for all the strengthened specimens. Fib Bulletin 90 (2019) predictions were more accurate but still overestimated the capacity of all specimens strengthened with short-length CFRP plates.

## Figures and Tables

**Figure 1 polymers-15-01621-f001:**
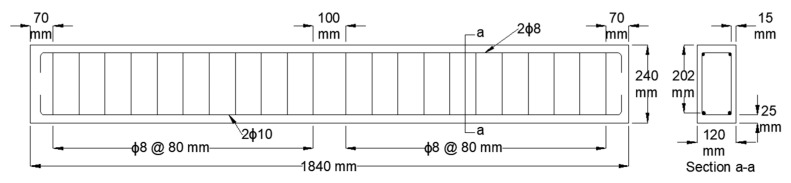
RC beam geometry and reinforcement details.

**Figure 2 polymers-15-01621-f002:**
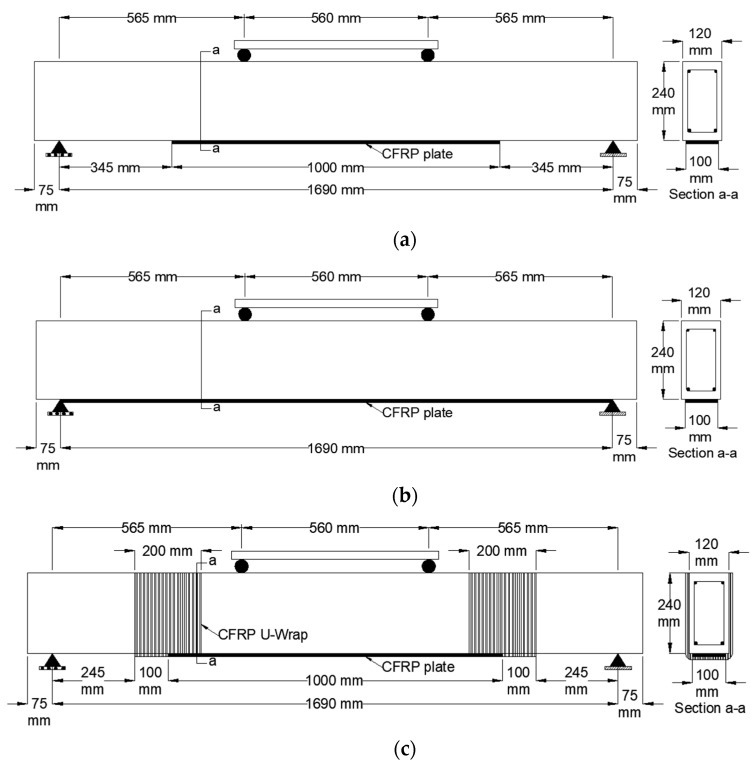
Strengthening details of specimens: (**a**) S; (**b**) F; (**c**) S-U.

**Figure 3 polymers-15-01621-f003:**
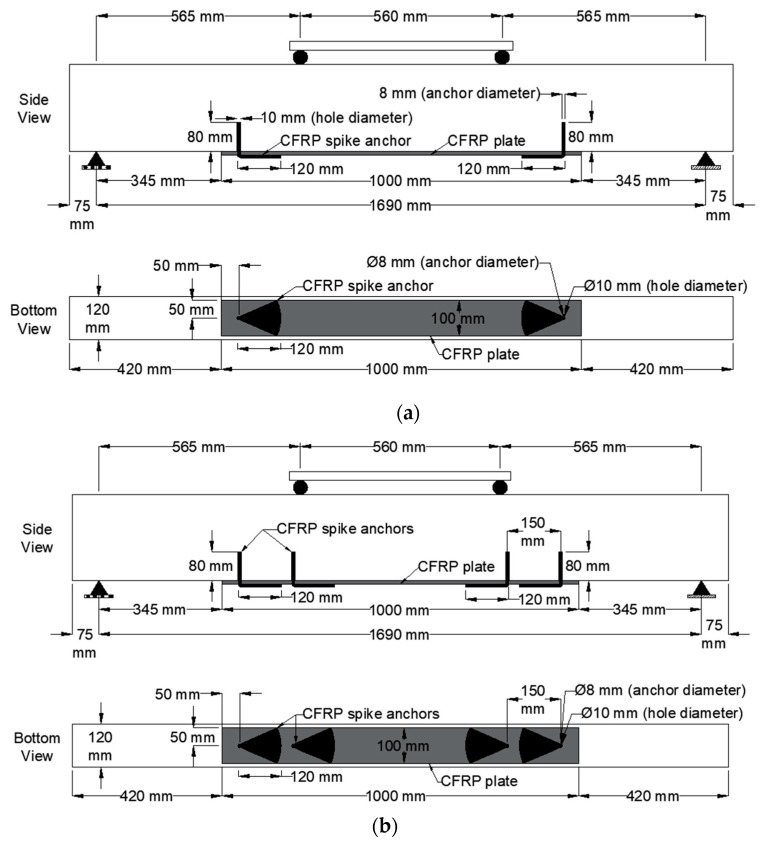
CFRP spike anchor details for specimens: (**a**) S-2A; (**b**) S-4A(d); (**c**) S-4A(s).

**Figure 4 polymers-15-01621-f004:**
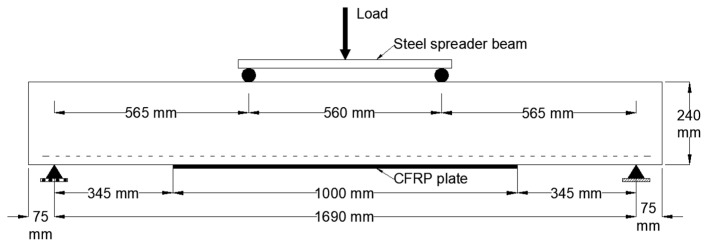
Test setup for S specimen as an example.

**Figure 5 polymers-15-01621-f005:**
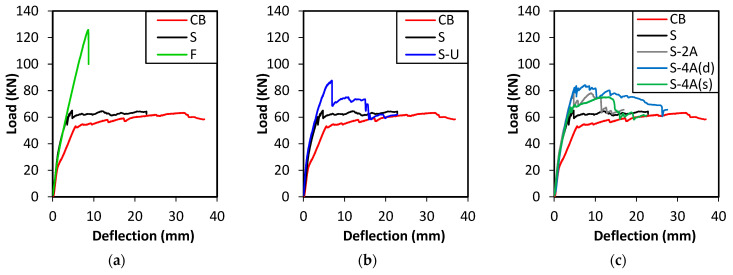
Load-deflection curves for specimens: (**a**) CB, S, F; (**b**) CB, S, S-U; (**c**) CB, S, S-2A, S-4A.

**Figure 6 polymers-15-01621-f006:**
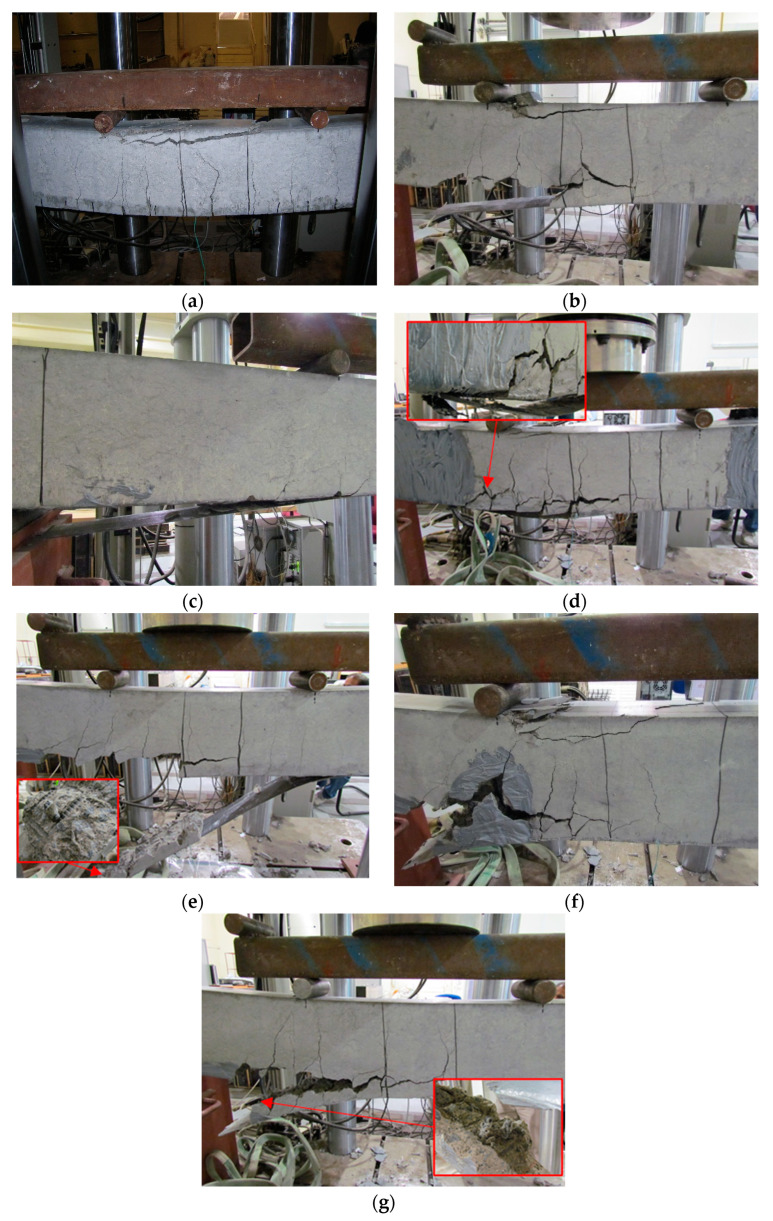
Failure modes of specimens: (**a**) CB; (**b**) S; (**c**) F; (**d**) S-U; (**e**) S-2A; (**f**) S-4A(d); (**g**) S-4A(s).

**Figure 7 polymers-15-01621-f007:**
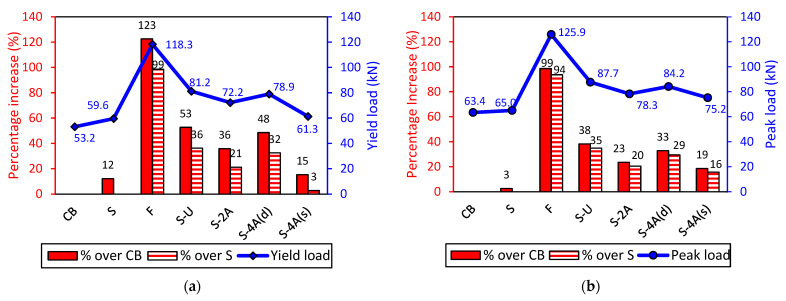
Flexural performance of the tested specimens: (**a**) Yield load; (**b**) Peak load.

**Figure 8 polymers-15-01621-f008:**
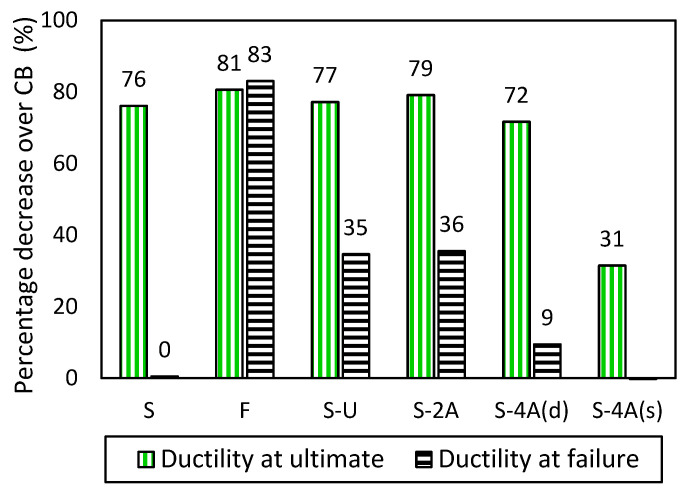
Percentage decrease in the ductility at ultimate and failure states with respect to the control beam.

**Table 1 polymers-15-01621-t001:** Details of the tested specimens.

Specimen	Details
CB	Control beam—unstrengthened
S	Beam strengthened with a 1 m long CFRP plate (short length)
F	Beam strengthened with a 1.69 m CFRP plate (full-length)
S-U	Beam strengthened with a 1 m long CFRP plate and anchored with end U-wraps
S-2A	Beam strengthened with a 1 m long CFRP plate and anchored with two CFRP spike anchors (one on each side)
S-4A(d)	Beam strengthened with a 1 m long CFRP plate and anchored with four distributed CFRP spike anchors
S-4A(s)	Beam strengthened with a 1 m long CFRP plate and anchored with four CFRP spike anchors (two on each side)

**Table 2 polymers-15-01621-t002:** Concrete mix proportions and mechanical properties.

Cement (kg/m^3^)	Micro Silica (kg/m^3^)	Sand (kg/m^3^)	Coarse Aggregate (kg/m^3^)	Additives (kg/m^3^)	w/c	Density (kg/m^3^)	Compressive Strength (MPa)	Tensile Strength (MPa)
430	20	998	842.6	9.45	0.36	2459	44.6 ± 1.7	4.3 ± 0.6

**Table 3 polymers-15-01621-t003:** Mechanical properties of CFRP sheet, CFRP plate, and epoxy adhesives.

Material	Material Name	Density	Thickness (mm)	Tensile Strength (MPa)	Elastic Modulus (MPa)	Strain at Break (%)
CFRP plate	Sika CarboDur	1.60 g/cm^3^	1.2	3100	170,000	1.80
CFRP sheet	SikaWrap-300C	1.82 g/cm^3^	0.167	4000	230,000	1.70
Epoxy adhesive	Sikadur-30 LP	1.65 kg/L		17	10,000	
Epoxy adhesive	Sikadur-330	1.40 kg/L		30	4500	0.90

**Table 4 polymers-15-01621-t004:** Summary of test results.

Specimen Designation	Rebar Yielding		Ultimate State		Failure		Failure Mode ^a^	Ductility ^b^	
	Load (kN)	Mid-Span Deflection (mm)	Load (kN)	Mid-Span Deflection (mm)	Load (kN)	Mid-Span Deflection (Mm)		At Ultimate	At Failure
CB	53.2	5.6	63.4	31.9	58.5	36.9	SY, CC	5.71	6.60
S	59.6	3.5	65.0	4.7	52.0	22.8	PD + C, CC	1.37	6.57
F	118.3	7.8	125.9	8.7	100.7	8.8	PD	1.11	1.12
S-U	81.2	5.3	87.7	6.9	70.1	22.8	LPD, RU	1.30	4.32
S-2A	72.2	4.0	78.3	4.7	62.6	16.9	PD + C, AP	1.19	4.26
S-4A(d)	78.9	4.6	84.2	7.5	67.3	27.5	PD + C, AP, CC	1.62	5.98
S-4A(s)	61.3	3.4	75.2	13.3	60.1	22.4	PD + C, AP	3.92	6.62

^a^ SY: steel yielding, CC: concrete crushing, PD + C: plate debonding with concrete cover, PD: plate debonding, LPD: local plate debonding, RU: rupture of CFRP U-wraps, AP: anchor pullout. ^b^ Ductility at ultimate is the ratio of mid-span deflection at ultimate load over mid-span deflection at yield load; Ductility at failure is the ratio of mid-span deflection at failure load over mid-span deflection at yield load.

**Table 5 polymers-15-01621-t005:** Summary of equations for determining flexural capacity of FRP-strengthened members.

Design Code	Equations
ACI440.2R-17 [49]	$M_{n}=A_{s}f_{y}\left(d-\frac{\beta_{1}c}{2}\right)+\Psi_{f}A_{f}f_{fe}\left(d_{f}-\frac{\beta_{1}c}{2}\right)$; $A_{f}=nt_{f}w_{f}$; $\varepsilon_{fd}=0.41\sqrt{\frac{f_{c}^{\prime }}{nE_{f}t_{f}}}\leq 0.9\varepsilon_{fu}$; $\varepsilon_{fe}=\varepsilon_{cu}\left(\frac{d_{f}-c}{c}\right)\leq \;\varepsilon_{fd}$ $\varepsilon_{c}=(\varepsilon_{fe})\left(\frac{c}{d_{f}-c}\right)\leq \varepsilon_{cu};\;\varepsilon_{s}=(\varepsilon_{fe})\left(\frac{d-c}{d_{f}-c}\right)$; $f_{s}=E_{s}\varepsilon_{s}\leq f_{y}$; $f_{fe}=E_{f}\varepsilon_{fe}$; $c=\frac{A_{s}f_{s}+A_{f}f_{fe}}{\alpha_{1}f_{c}^{\prime }\beta_{1}b}$ If concrete crushing controls: $\beta_{1}=\left\{\begin{array}{c} 0.85\;for\;f_{c}^{\prime }\leq 28\;\mathrm{MPa}\\ 0.85-0.00714\left(f_{c}^{\prime }-28\right)\;for\;28<f_{c}^{\prime }<56\;\mathrm{MPa};\;\alpha_{1}=0.85\\ 0.65\;for\;f_{c}^{\prime }\geq 56\;\mathrm{MPa} \end{array}\right.$ $;\;\alpha_{1}=0.85$ If FRP debonding controls: $\beta_{1}=\frac{4\varepsilon_{c-}^{\prime }\varepsilon_{c}}{6\varepsilon_{c-}^{\prime }2\varepsilon_{c}}$ $\;;\;\alpha_{1}=\frac{3\varepsilon_{c}^{\prime }\varepsilon_{c}-\varepsilon_{c}^{2}}{3\;\beta_{1}\varepsilon_{c}^{\prime }{}^{2}}$ $;\;\varepsilon_{c}^{\prime }=\frac{1.7\;f_{c}^{\prime }}{E_{c}}$ $;\;E_{c}=4700\surd f_{c}^{\prime }$
CSA-S806.12 (R 2017) [57]	$M_{n}=A_{s}f_{y}\left(d-\frac{\beta_{1}c}{2}\right)+A_{F}f_{F}\left(d_{f}-\frac{\beta_{1}c}{2}\right)$$;\;A_{f}=nt_{f}w_{f}$; $\varepsilon_{Fmax}=0.41\sqrt{\frac{f_{c}^{\prime }}{n_{f}E_{f}t_{f}}}\leq 0.007;$$\varepsilon_{F}=0.0035\left(\frac{d_{f}-c}{c}\right)\leq \;\varepsilon_{Fmax}$ $\varepsilon_{c}=(\varepsilon_{F})\left(\frac{c}{d_{f}-c}\right);\;\varepsilon_{s}=(\varepsilon_{F})\left(\frac{d-c}{d_{f}-c}\right);\;$$f_{s}=E_{s}\varepsilon_{s}\leq f_{y}$$;\;f_{F}=E_{F}\varepsilon_{F}$; $c=\frac{A_{s}f_{s}+A_{F}f_{F}}{\alpha_{1}f_{c}^{\prime }\beta_{1}b}$; If $\left(\frac{c}{d}\right)\geq \frac{7}{7+2000\varepsilon_{Fu}}$: $\alpha_{1}=0.85-0.0015f_{c}^{\prime }\;\geq 0.67$ $\beta_{1}=0.97-0.0025f_{c}^{\prime }\;\geq 0.67$
fib Bulletin 90 (2019) [58]	$M_{Rd}=A_{s1\;}\sigma_{s1\;}\left(d_{s1}-k_{2}x\right)+A_{f\;}E_{f}\varepsilon_{f}\left(h-k_{2}x\right)+A_{s2\;}\sigma_{s2\;}\left(k_{2}x-d_{s2}\right)$; $\sigma_{s1\;}=\min\left(\varepsilon_{c}\frac{d_{s1}-x}{x},\;\frac{f_{yd}}{E_{s}}\right)E_{s}$$;\;\sigma_{s2\;}=\min\left(\varepsilon_{c}\frac{x-d_{s2}}{x},\;\frac{f_{yd}}{E_{s}}\right)E_{s}$; $\varepsilon_{f}=\varepsilon_{c}\;\frac{h-x}{x}$$\leq \;\varepsilon_{fbd,IC}$ $k_{1}=\left\{\begin{array}{c} 1000\varepsilon_{c}\left(0.5-\frac{1000}{12}\varepsilon_{c}\right)\;for\;\varepsilon_{c}\;\leq 0.002\\ 1-\frac{2}{3000}\varepsilon_{c}\;for\;0.002\leq \varepsilon_{c}\leq 0.0035 \end{array}\right.$ $k_{2}=\left\{\begin{array}{c} \frac{8-1000\varepsilon_{c}}{4\left(6-1000\varepsilon_{c}\right)}\;for\;\varepsilon_{c}\;\leq 0.002\\ \frac{1000\varepsilon_{c}\left(3000\varepsilon_{c}-4\right)+2}{2000\varepsilon_{c}\left(3000\varepsilon_{c}-2\right)}\;for\;0.002\leq \varepsilon_{c}\leq 0.0035 \end{array}\right.$ Using the simplified stress method: $\varepsilon_{fbd,IC}=\frac{f_{bd.IC}}{E_{f}}$ $;\;f_{bd.IC}=\frac{k_{cr,k}\;k_{k}k_{b}\sqrt{\frac{2E_{f}}{t_{f}}f_{cm}^{\frac{2}{3}}}}{\gamma_{fb}}$ $;\;k_{cr}=1.8$ $;\;k_{k}=0.17$ $;\;\gamma_{fb}=1.5$ $k_{b}=\sqrt{\frac{2-\frac{b_{f}}{b}}{1+\frac{b_{f}}{b}}}\geq 1$

Note: where $M_{n}\mathrm{and}\;M_{Rd}$: nominal flexural moment capacity (Nmm); $A_{s}$: area of steel reinforcement (mm^2^);
$f_{y}$: tensile yield strength of the steel bars (MPa); d: effective depth of the steel reinforcement (mm); $\beta_{1}:$ ratio of the depth of equivalent rectangular stress block to the depth of the neutral axis; c: distance from the extreme compression fiber to the neutral axis (mm);
n: number of FRP plies; $t_{f}$: thickness of FRP ply (mm); $w_{f}$: width of FRP sheet (mm); $A_{f}\;\mathrm{and}\;A_{F}$: area of FRP reinforcement (mm^2^);
$\Psi_{f}$: strength reduction factor = 0.85; $f_{fe}\;\mathrm{and}\;f_{F}$: effective stress in the FRP (MPa); $E_{f}\;\mathrm{and}\;E_{F}$: elastic modulus of FRP (MPa); $\varepsilon_{fe}$: effective strain in the FRP (mm/mm); $d_{f}$: effective depth of FRP reinforcement (mm); $\varepsilon_{fd}:$ debonding strain of the FRP (mm/mm); $\varepsilon_{fu}\;\mathrm{and}\;\varepsilon_{Fu}$: rupture strain of the FRP (mm/mm); $f_{c}^{\prime }$: compressive strength of concrete (MPa); $\varepsilon_{cu}$: ultimate axial strain of unconfined concrete = 0.003; $\varepsilon_{c}$: strain in concrete (mm/mm); $\varepsilon_{s}$: strain in steel reinforcement (mm/mm); $f_{s}:$ stress in the steel reinforcement (MPa); $f_{y}$ and $f_{yd}$: yield stress of steel (MPa); $E_{s}$: elastic modulus of steel (MPa); $\alpha_{1}$: multiplier on $f_{c}^{\prime }$ to determine intensity of an equivalent rectangular stress distribution for concrete; $\varepsilon_{c}^{\prime }$: compressive strain of unconfined concrete corresponding to $f_{c}^{\prime }$ (mm/mm); $E_{c}$: elastic modulus of concrete (MPa); $\varepsilon_{Fmax}$: maximum tensile strain in the FRP (mm/mm); $\varepsilon_{F}$: strain in the FRP reinforcement (mm/mm); $A_{s1}$: area of bottom steel reinforcement (mm^2^); $A_{s2}$: area of top steel reinforcement (mm^2^); $\sigma_{s1}$: stress in the bottom steel reinforcement (MPa); $\sigma_{s2}$: stress in the top steel reinforcement (MPa); $d_{s1}$: distance from extreme compression fiber to centroid of tension reinforcement (mm); $d_{s2}$: distance from extreme compression fiber to centroid of compression reinforcement (mm); h: overall height of the member (mm); x: depth of the neutral axis (mm); $k_{1}$ and $k_{2}$: factors for the average compressive stress and the lever arm of the parabolic–rectangular stress distribution of concrete; $\varepsilon_{fbd,IC}$: design strain in the FRP reinforcement (mm/mm); $f_{bd.IC}$: design stress in the FRP reinforcement (MPa); $k_{cr,k}$: numerical coefficient = 1.8; $k_{k}:$ numerical doefficient = 0.17; $k_{b}$: shape factor; $f_{cm}$: compressive strength of concrete (MPa); $\gamma_{fb}$: partial factor = 1.5; $b_{f}$: width of FRP reinforcement (mm); b: width of member (mm).

**Table 6 polymers-15-01621-t006:** Ultimate load predictions.

Specimen	*P_exp_* (kN)	*P_pred_* (kN)			*P_exp_/P_pred_*		
		ACI440.2R-17	CSA-S806.12 (R 2017)	fib Bulletin 90 (2019)	ACI440.2R-17	CSA-S806.12 (R 2017)	fib Bulletin 90 (2019)
S	65.0	156.0	154.1	107.2	0.42	0.42	0.61
F	125.9	156.0	154.1	107.2	0.81	0.82	1.17
S-U	87.7	156.0	154.1	107.2	0.56	0.57	0.82
S-2A	78.3	156.0	154.1	107.2	0.50	0.51	0.73
S-4A(d)	84.2	156.0	154.1	107.2	0.54	0.55	0.79
S-4A(s)	75.2	156.0	154.1	107.2	0.48	0.49	0.70
				Average	0.55	0.56	0.80

## Data Availability

Data are contained within the article.

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
