# Peer review of "CFRP U-Wraps and Spike Anchors for Enhancing the Flexural Performance of CFRP-Plated RC Beams"

_polymers, 2023, doi:10.3390/polym15071621_

Round 1
Reviewer 1 Report
Figure 4 for test setup, this figure is for type S specimen with FRP. You can delete FRP or add "S specimens for example" for showing the test setup.
The ductilities of strengthened beams are lower than that of the control specimen. That is strange. What is the response if continuing to apply loading after the ultimate points of load-deformation curves of strengthened specimens? When reading the paper "Effect of defects in externally bonded FRP reinforced concrete", after FRP debonding, if continue to apply the load, the final ductility will be similar to the control specimens.
When writing the Introduction, please include the grooving anchoring system: "Epoxy interlocking: A novel approach to enhance FRP-to-concrete bond behavior" and "RC members externally strengthened with FRP composites by grooving methods including EBROG and EBRIG: A state-of-the-art review".
Author Response
A response file has been attached.

Reviewer 2 Report
This paper presents experimental study on RC beams strengthened with FRP with different forms of anchorage. The experimental program and results are well presented, and the paper is of interests to the readers. The authors could enhance the paper based on the following comments.
1. Please check 'A total of sixteen standard concrete cylinders of length of 2000 mm and diameter of 100 mm'. It seems that the length should be 200 mm.
2. The sources of the FRP material properties should be given.
3. The resolution of Figure 6 can be enhanced, the figures in the current form are vague.
4. Please explain why the strength of Beam S-4Ad is larger than Beam S-4As.
5. The debonding of FRP plates are quite normal in practice, while the ends of the FRP plate are not easy to be anchoraged due to the presence of the beam-column joints. Please discuss if there is another form of FRP instead of anchorage (e.g. FRP grid, Eng Struct. 2022 272 115020) that could be well bonded to concrete.
6. Ductility should be defined.
7. Please double-check the results in Table 6. The predicted values are considerablely overestimated.
Author Response
A response file has been attached.

Round 2
Reviewer 2 Report
The authors have addressed the comments properly.